# Liver stage *P. falciparum* antigens highly targeted by CD4+ T cells in malaria-exposed Ugandan children

Gonzalo R. Acevedo[1], Sophie S. Samiee[1], Mikias Ilala[1], Justine Levan[1], Meagan E. Olive[1], Riana D. Hunter[1], Mary Prahl[2], Raja Rajalingam[3], John Rek[4¤], Grant Dorsey[1], Margaret E. Feeney[1,2]*

**1** Department of Medicine, University of California San Francisco, San Francisco, California, United States of America, **2** Department of Pediatrics, University of California San Francisco, San Francisco, California, United States of America, **3** Department of Surgery, University of California San Francisco, San Francisco, California, United States of America, **4** Infectious Diseases Research Collaboration, Kampala, Uganda

¤ Current address: Adaptive Malaria Control Program for Uganda - NMCD, Ministry of Health, Kampala, Uganda
* margaret.feeney@ucsf.edu

**Data availability statement:** All relevant data are within the manuscript and its Supporting Information files, or were sourced from the public databases PlasmoDB (https://www.plasmodb.org; dataset IDs: DS_ee861a9187, DS_a2967e5664, DS_66f9e70b8a, DS_0c4bd1fca3, DS_715bf2deda, DS_7b1bac6cd1, DS_49781e8c33, DS_cf291e2adb, DS_337c46b0f0) and the

## Abstract

T cell responses against liver stage *Plasmodium* help protect against reinfection, but the antigens and epitopes targeted by these T cells are largely unknown. This knowledge gap has impeded mechanistic studies to identify the effector functions most critical for protection. We performed a bioinformatic analysis of gene expression datasets to identify plasmodial genes that are highly and selectively expressed during liver stage infection and predict epitopes within them likely to bind MHC-II molecules prevalent in Uganda. We then tested their recognition by malaria-exposed Ugandan children. In over two-thirds of the children, we detected a peripheral blood CD4+ T cell response to one or more antigens. The most highly targeted antigen, LISP1, contained several epitopes, including one that was promiscuously presented and recognized by most participants. These novel liver stage *P. falciparum* epitopes should be explored as potential vaccine targets and will facilitate the development of tools to interrogate antimalarial immunity at the single-cell level and inform future vaccine development efforts.

## Author summary

Malaria causes over half a million deaths every year, over 80% of which are accounted for by children under five years of age. The currently available vaccines to prevent this disease provide limited, short-lived protection. Prior evidence suggests that generating immunity against the liver stage of *P. falciparum* is a promising strategy to improve vaccine efficacy and durability. The antigens expressed by liver stage parasites that are targeted by immune cells are still mostly unidentified. In this research, we took advantage of previously published *Plasmodium* gene expression datasets and bioinformatic tools to predict T cell epitopes in parasite proteins of selective liver stage expression. Using specimens from Ugandan children living in a high malaria transmission area, we confirmed that several of these epitopes are in fact targets of CD4+ T cells in naturally developed immune

MalariaGEN *Plasmodium falciparum* Community Project (https://www.malariagen. net/; dataset "Catalogue of Genetic Variation in *P. falciparum* - v6.0", as described in 'An open dataset of *Plasmodium falciparum* genome variation in 7,000 worldwide samples', https:// doi.org/10.12688/wellcomeopenres.16168.1)

**Funding:** This work was funded by the National Institutes of Health (grant numbers 5R01AI093615 and K24AI113002 to MEF; and U19AI089674 to GD and MEF) and the National Center for Advancing Translational Sciences, National Institutes of Health, through UCSF-CTSI (grant number UL1 TR001872 to GRA). The funders had no role in study design, data collection and analysis, decision to publish, or preparation of the manuscript

**Competing interests:** The authors have declared that no competing interests exist

responses. One particular protein, LISP1, was identified as a frequently targeted antigen among children in this study. Our results generate valuable tools to examine immunity against liver stage malaria and inform the development of improved vaccines.

## Introduction

Malaria remains among the deadliest infectious diseases worldwide [1,2]. Although the recent approval of the RTS,S and R21 vaccines represents a welcome advance in malaria prevention, substantial concerns remain about the modest level and limited duration of protection afforded by these first-generation vaccines [3,4]. Both are based on a single allele of a single malaria antigen (CSP), and the efficacy of RTS,S against unmatched strains (which comprise >90% of global infections) is significantly lower than against matched strains [5]. Therefore, it is clear that new vaccines capable of inducing durable protection in highly malaria-exposed populations are much needed, and it is widely acknowledged that such next-generation vaccines should incorporate additional *P. falciparum* antigens.

An effective T cell response against liver stage *Plasmodium* infection can generate long-lasting, strain-transcendent protection against reinfection. The most successful experimental strategies of malaria immunization to date involve inoculation with live, attenuated sporozoites [6]. Vaccination with radiation-attenuated sporozoites (RAS), which infect hepatocytes but arrest development at the liver stage, confers almost complete protection against homologous challenge and displays high efficacy against field strains following natural exposure [6]. Mechanistic studies in humans, non-human primates, and mice indicate that T cells are critical for the protection induced by RAS vaccination [7–10] and that, at least in mice, CD4+ T cell help is necessary for establishing a long-lived CD8+ T cell response in the liver [11,12]. Studies of experimental inoculation with genetically attenuated sporozoites (GAS) [13–16] and non-attenuated sporozoites under chemoprophylaxis [17,18] also support a critical role for liver stage T cell responses in immunity against malaria. Importantly, the stage of liver development at which the parasite undergoes developmental arrest strongly influences immune protection, with late liver stage-arresting infections engendering stronger protection than early-arresting ones [14,19–21]. This suggests that T cells primed during the late intrahepatic development of the parasite are crucial for immune protection.

Despite the compelling evidence that liver stage-specific T cells are critical for protection, the precise antigens targeted by these T cells are almost entirely unknown. Very few liver stage *P. falciparum* CD4+ T cell epitopes have been identified and characterized in humans [22]. This critical gap has hindered efforts to identify correlates of protective immunity to natural malaria exposure. Many antigens traditionally classified as pre-erythrocytic are abundantly transcribed in asexual blood parasites or rapidly downregulated in the transition from sporozoites to liver stage forms. Therefore, T cells responding against them may be primed in sites other than the liver, limiting the utility of these antigens in the characterization of liver stage-specific T cells and their role in protection. The lack of validated liver stage antigens poses a major hurdle to our understanding of anti-malarial immunity and the evaluation of vaccine candidates.

In this report, we leverage pre-existing gene expression data from malaria parasites to identify candidate CD4+ T cell epitopes in *P. falciparum* antigens that are primarily expressed during the liver stage of development. Then, using specimens from an observational study in a high malaria transmission area in Uganda (PRISM) [23], we assessed the recognition of these candidates by CD4+ T cells from exposed children. We detected CD4+ T cell responses in two-thirds of the participants, confirming the T cell immunogenicity of antigens identified through this pipeline and highlighting LISP1 as a frequently targeted antigen containing

several broadly recognized epitopes. Further, we analyzed potential HLA restriction elements of some of these epitopes, which we confirmed using peptide-MHC multimers. These novel tools will enable a more comprehensive characterization of human T cell immunity against liver stage *P. falciparum*, its role in protection, and the mechanisms that hinder its efficacy.

## Results

### Bioinformatic selection of liver stage *P. falciparum* candidate antigens

First, we sought to identify genes selectively expressed during the liver stage of *P. falciparum*. In the absence of published *P. falciparum*-specific genome-wide expression data, we adopted a strategy that leveraged transcriptomic datasets from the related species *P. vivax* [24] and *P. cynomolgi* [25]. These two datasets were merged with data representing gene expression in the asexual intraerythrocytic cycle of *P. falciparum*, *P. vivax*, *P. cynomolgi*, and *P. berghei* to generate a master dataset (**Fig 1A**), as detailed in the Methods section. Seeking to increase the likelihood of selecting candidates with conserved regulation of gene expression across *Plasmodium* species, we hypothesized that genes with an evolutionarily preserved genomic context would likely have higher conservation in their regulatory elements and therefore more conserved expression dynamics as well. Hence, the data was filtered to only retain genes with syntenic orthology across species. The final dataset comprised 3848 genes across the life cycle of four *Plasmodium* species.

We prioritized candidate antigens on the basis of their gene expression pattern by creating a scoring function to reflect the following criteria: (i) selective liver stage expression in both *P. vivax* and *P. cynomolgi* (**Fig 1B**); (ii) low expression during the intraerythrocytic development

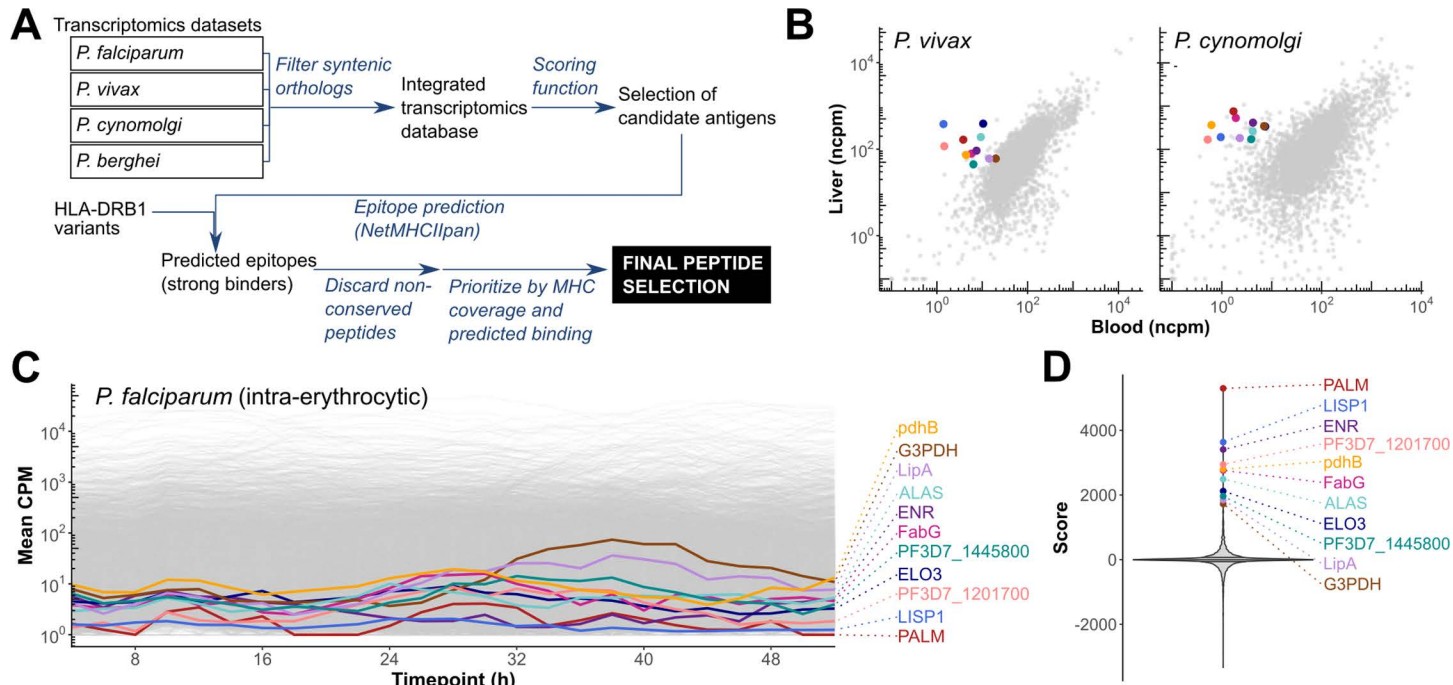

**Fig 1. Selection of candidate liver stage *P. falciparum* antigens. A.** Overview of antigen selection and epitope prediction strategy. **B.** Secondary analysis of transcription for genes with syntenic orthology to *P. falciparum* in the liver and blood stages of *P. vivax* and *P. cynomolgi*. Each point corresponds to a gene. The selected candidate antigens are highlighted. Colors match across panels **B**–**D** for each gene. **C.** Secondary analysis of the expression of syntenic ortholog genes in *P. falciparum* across the intraerythrocytic asexual development cycle. **D.** Distribution of the scores calculated for each gene. ncpm: normalized counts per million reads.

cycle for *P. falciparum* (**Fig 1C**); (iii) strongly correlated expression at matched life cycle stages between *P. falciparum* and other *Plasmodium* species. (Details of the scoring function used to select candidate antigens are provided in the Methods section). Among the top-scoring genes, 11 were selected as candidate antigens (**Fig 1D** and **Table 1**). Predominant liver stage expression of these candidate genes in *P. falciparum* was corroborated in a recent dataset reporting gene expression for *P. falciparum* sporozoites and intrahepatocytic parasites in a human liver-chimeric mouse model [26] (S1 Fig).

## Epitope prediction and peptide design

Next, we sought to identify peptides within these 11 candidate liver stage antigens that would likely bind class II MHC molecules common within the population at our study site in Tororo, Uganda [23]. We focused on the three HLA-DRB1 alleles with the highest prevalence in the PRISM study cohort: DRB1*11:01, *13:02, and *15:03, each of which is expressed by >25% of individuals in the cohort [27]. Overall, 71.9% of cohort participants expressed at least one of these three DRB1 alleles (n = 272, **Fig 2A**). We used NetMHCIIpan [28] to identify 15-mers derived from the reference (3D7) amino acid sequences of candidate antigens [29,30] that were predicted to bind to our target MHC. Out of 8407 unique 15-mers, 295 were predicted to strongly bind at least one allele of interest (Rank% score ≤ 0.5).

Predicted epitopes that were contiguous and shared 11 or more residues were merged into a single peptide, to a maximum length of 19 amino acids. Forty-one peptides were shortlisted for synthesis as 15- to 19-mers. Two peptides (ALAS$_{[474–492]}$ and ALAS$_{[475–493]}$) were excluded from the study upon observation of non-specific responses among malaria-unexposed donors in preliminary tests. A final set of 39 peptides was selected for further testing (**Fig 2B** and **2C**). These were predicted to provide broad coverage of the three target alleles, with some binding promiscuity (**Fig 2D**).

## CD4⁺ T cell responses to liver stage antigens are detectable in most malaria-exposed children

To determine whether our predicted epitopes were in fact recognized by CD4⁺ T cells, we screened 34 children 5–11 years of age who were enrolled in a longitudinal observational study of childhood malaria in the Tororo District of eastern Uganda, a region of intense

**Table 1. Gene IDs in reference strains of *Plasmodium* species for syntenic ortholog genes selected as candidate antigens.**

| Antigen | Gene ID | | |
|---|---|---|---|
| | *P. falciparum* | *P. vivax* | *P. cynomolgi* |
| ALAS | PF3D7_1246100 | PVP01_1463100 | PcyM_1469400 |
| ELO3 | PF3D7_0920000 | PVP01_0718400 | PcyM_0718500 |
| ENR | PF3D7_0615100 | PVP01_1134000 | PcyM_1135600 |
| FabG | PF3D7_0922900 | PVP01_0721400 | PcyM_0721700 |
| G3PDH | PF3D7_1114800 | PVP01_0915500 | PcyM_0917100 |
| LipA | PF3D7_1344600 | PVP01_1210200 | PcyM_1211900 |
| LISP1 | PF3D7_1418100 | PVP01_1330800 | PcyM_1336200 |
| PALM | PF3D7_0602300 | PVP01_1146600 | PcyM_1149900 |
| pdhB | PF3D7_1446400 | PVP01_1260600 | PcyM_1269400 |
| PF3D7_1201700 | PF3D7_1201700 | PVP01_1301000 | PcyM_1301500 |
| PF3D7_1445800 | PF3D7_1445800 | PVP01_1261200 | PcyM_1270000 |

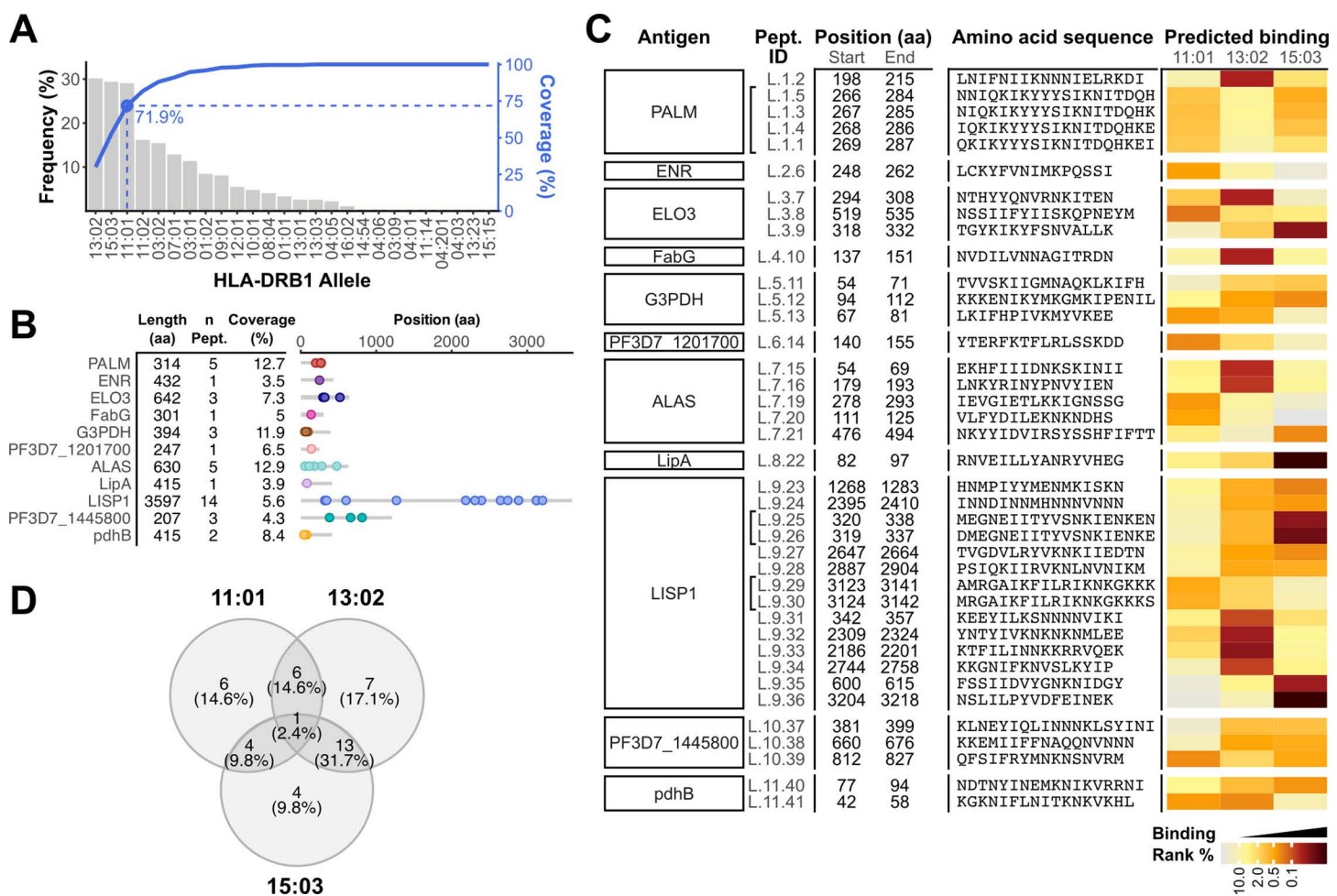

**Fig 2. Epitope prediction in selected liver stage *P. falciparum* antigens. A.** Prevalence (bars, left vertical axis) of DRB1 alleles (horizontal axis) in the study site was used to select targets for MHC binding prediction. 71.9% of the typed individuals express at least one of the three most prevalent alleles (blue, right vertical axis). **B.** Antigens chosen for epitope prediction, with their size, number of peptides selected for in vitro testing, percentage of the antigen sequence covered by them, and their position in the antigen. **C.** Peptides selected for in vitro testing, with their position in the antigen, amino acid sequence, and predicted binding to target HLA-DRB1 MHC. Brackets join IDs of 19-mers covering overlapping segments of the antigen. Heatmap represents the Rank% score from the prediction output. **D.** Overlap in predicted binding (Rank% ≤ 2.0) of peptides in **B** to targeted alleles. Numbers indicate the peptides predicted to bind to each allele, percentages represent their proportion over the total set of 39 peptides.

perennial *P. falciparum* transmission [23]. We selected individuals who expressed at least one of the three most prevalent HLA-DRB1 alleles (DRB1*11:01, *13:02, and *15:03) used in our epitope prediction. To maximize the likelihood of detecting liver stage-specific cells in the peripheral circulation, we selected specimens collected within three months of a donor's last recorded episode of malaria, based on published human and primate data indicating that CD4+ T cell responses peak in the early months after infection [7,31].

Recognition of candidate antigens was assessed by IFN-γ ELISPOT. Since responding cell frequencies were low in pilot studies, we implemented a short-term *in vitro* expansion step (consisting of stimulation with a pool containing all 39 peptides) in order to increase the sensitivity of detection, as outlined in **Fig 3A**. This protocol selectively expanded peptide-specific CD4+ T cells from PBMC without increasing the assay background (S2 Fig). Cells were tested by IFN-γ ELISPOT on day 14 for response against the total pool. Among the 34 total donors

tested, a response to the peptide pool was observed in 23 participants (67.6%, **Fig 3B**). No responses were observed among malaria-naïve adult control PBMC that were expanded and tested in parallel. In our Ugandan study participants, responses ranged greatly in magnitude, from 240 to 5580 spot-forming units per million cells (IQR: 680-3180 SFUPMC). In 18 of the 23 cases with a detected response, cell growth was sufficient to enable further characterization by flow cytometry. Intracellular cytokine staining assays confirmed that the IFN-γ-producing cells were predominantly CD4⁺ T cells (S2B Fig), with up to 51.2% of CD4⁺ T cells responding against the peptide pool. The frequency of IFN-γ-producing CD4⁺ T cells observed by intracellular cytokine staining strongly correlated with the response measured by ELISPOT (**Fig 3C**). Responses were notably more frequent among those sampled at 2–12 weeks

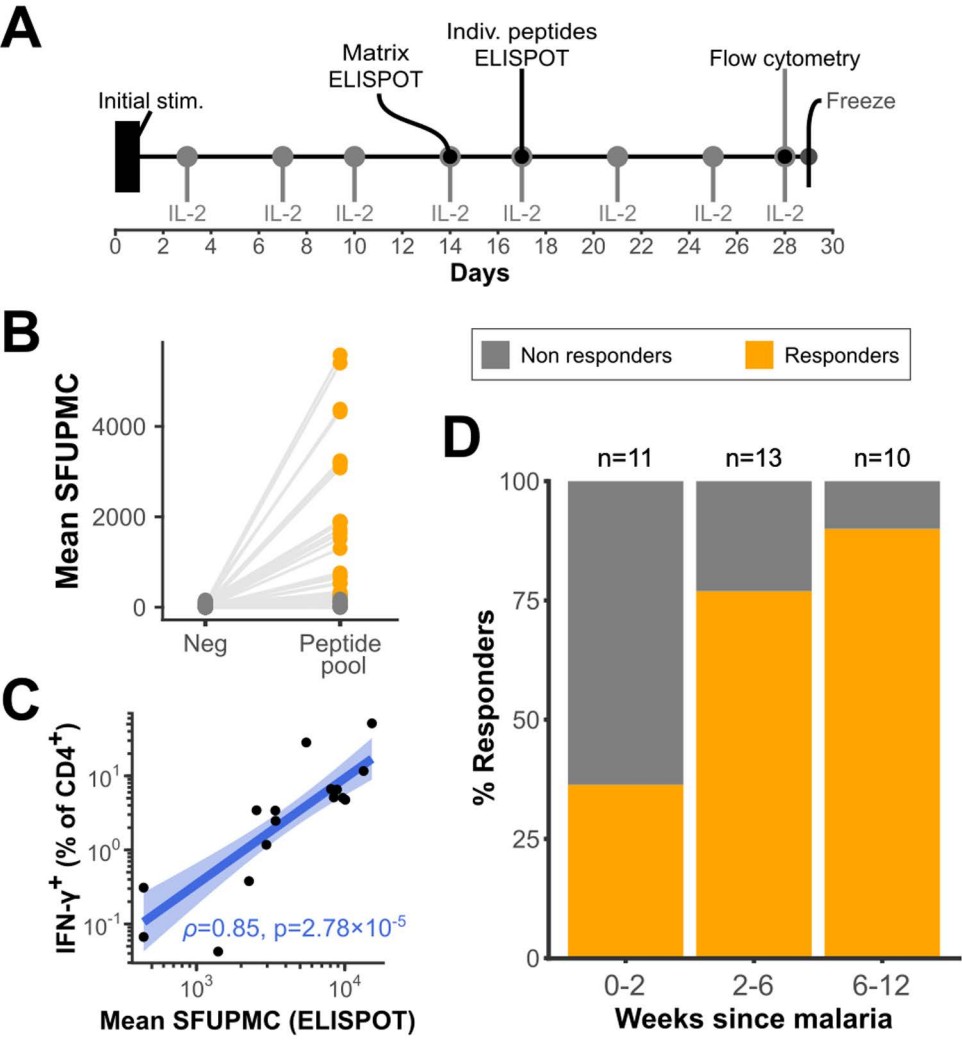

**Fig 3. CD4⁺ T cells from malaria-exposed children respond against pooled candidate epitopes in vitro. A.** Protocol for in vitro T cell expansion and testing. **B.** Response against the 39-peptide pool measured by IFN-γ ELISPOT on day 14 of T cell expansion. Each point represents a T cell line generated from an individual donor using the protocol outlined in A. **C.** Correlation between total IFN-γ-producing cells measured by ELISPOT and the frequency of IFN-γ⁺ CD4⁺ T cells measured by flow cytometry in response to the 39-peptide pool. Robust linear regression fit of the data, with Spearman's $\rho$ and p-value, shown in blue. **D.** Proportion of donors responding to pooled peptides, grouped by recency of their last malaria episode. SFUPMC: spot-forming units per 10⁶ input cells.

post-infection (82.6%) than in those sampled within 2 weeks of their last malaria episode (36.4%; **Fig 3D**), indicating that it may take weeks for newly primed CD4+ T cells to enter the peripheral circulation.

## LISP1 is a prevalent target of malaria-specific CD4+ T cells

Responses to individual liver stage peptides were assessed by a two-stage matrix IFN-γ ELISPOT screening approach [32] (S3A–C Fig). Of the 11 candidate antigens tested, 10 were recognized by at least one study participant (**Fig 4A**). LISP1 was by far the most frequently recognized antigen, with responses in 64.7% of children (22/34). LISP1 is a protein important for the egress of parasites from the liver [33] and is abundantly expressed throughout the liver stage [33–36]. Other highly targeted antigens included PALM (6/34 responders, 17.6%) and the hypothetical protein encoded by PF3D7_1445800 (5/34, 14.7%). We identified nine peptides that were recognized by >10% of study participants, including seven in LISP1 and one in each of PALM and PF3D7_1445800. Several participants exhibited broad responses that targeted more than one antigen or multiple non-overlapping peptides within the same antigen (**Fig 4A**). Flow cytometry confirmed the responses to be mediated by CD4+ T cells, with a strong correlation between IFN-γ production as measured by ELISPOT and by flow cytometry (S3D Fig). The frequency of responding CD4+ T cells ranged widely, with the most highly enriched lines (targeting LISP1 peptides L.9.29 and L.9.30) exceeding 40% specificity (**Fig 4B**).

For some broadly targeted peptides, there was a clear association with a specific DRB1 allele, whereas others appeared to be promiscuously presented (**Fig 4A**). For instance, recognition of LISP1 peptide L.9.27 was associated with allele DRB1*15:03 (RATE tool [37] p = 0.017), and recognition of LISP1 peptide L.9.28 was highly associated with allele DRB1*13:02 (p = 0.007). In contrast, the overlapping peptides L.9.29/L.9.30 were recognized by HLA-diverse individuals (RATE tool p > 0.2 for all class II alleles), suggesting the presence of an epitope that is promiscuously presented or multiple overlapping epitopes. Correspondence of inferred restrictions with NetMHCIIpan-predicted binding was only moderate, as L.9.27 and L.9.28 were predicted to have similarly high probabilities of binding to both DRB1*13:02 and DRB1*15:03 (**Fig 2C**).

Based on these inferred HLA associations, we designed peptide-MHC (pMHC) tetramers to validate the presentation of epitope L.9.30 via DRB1*11:01, *13:02, and *15:03, and of L.9.27 via DRB1*15:03. By combining tetramer staining with intracellular IFN-γ staining, we were able to experimentally confirm that the epitope(s) in L.9.30 are presented via the DRB1*11:01 and *13:02 MHC variants, and that L.9.27 is presented via DRB1*15:03 (**Figs 4C** and S4). While T cell line 3059 was derived from a DRB1*13:02-homozygous donor, a large proportion of its L.9.30-specific (IFN-γ+) cells were not stained with the corresponding tetramer. This could suggest that additional, non-DRB1 class II molecules may present an epitope within L.9.30, further underscoring its HLA promiscuity.

## Discussion

A growing body of evidence indicates that malaria-specific CD4+ T cells respond against pre-erythrocytic forms of *P. falciparum*, and their functionality correlates with protection from reinfection in human studies [18,31]. In animal models of immunization with liver stage-arresting parasites, CD4+ T cells are required for sterilizing protection against reinfection, and liver-resident parasite-specific memory CD8+ T cell populations dwindle quickly in the absence of CD4+ help [11,12]. The unknown identity of antigens targeted by *Plasmodium*-specific T cells during the liver stage of infection has impeded our understanding of pre-erythrocytic immunity. Here, we combined analysis of stage-specific plasmodial gene expression data, MHC binding prediction, and *in vitro* assays of T cell recognition to identify

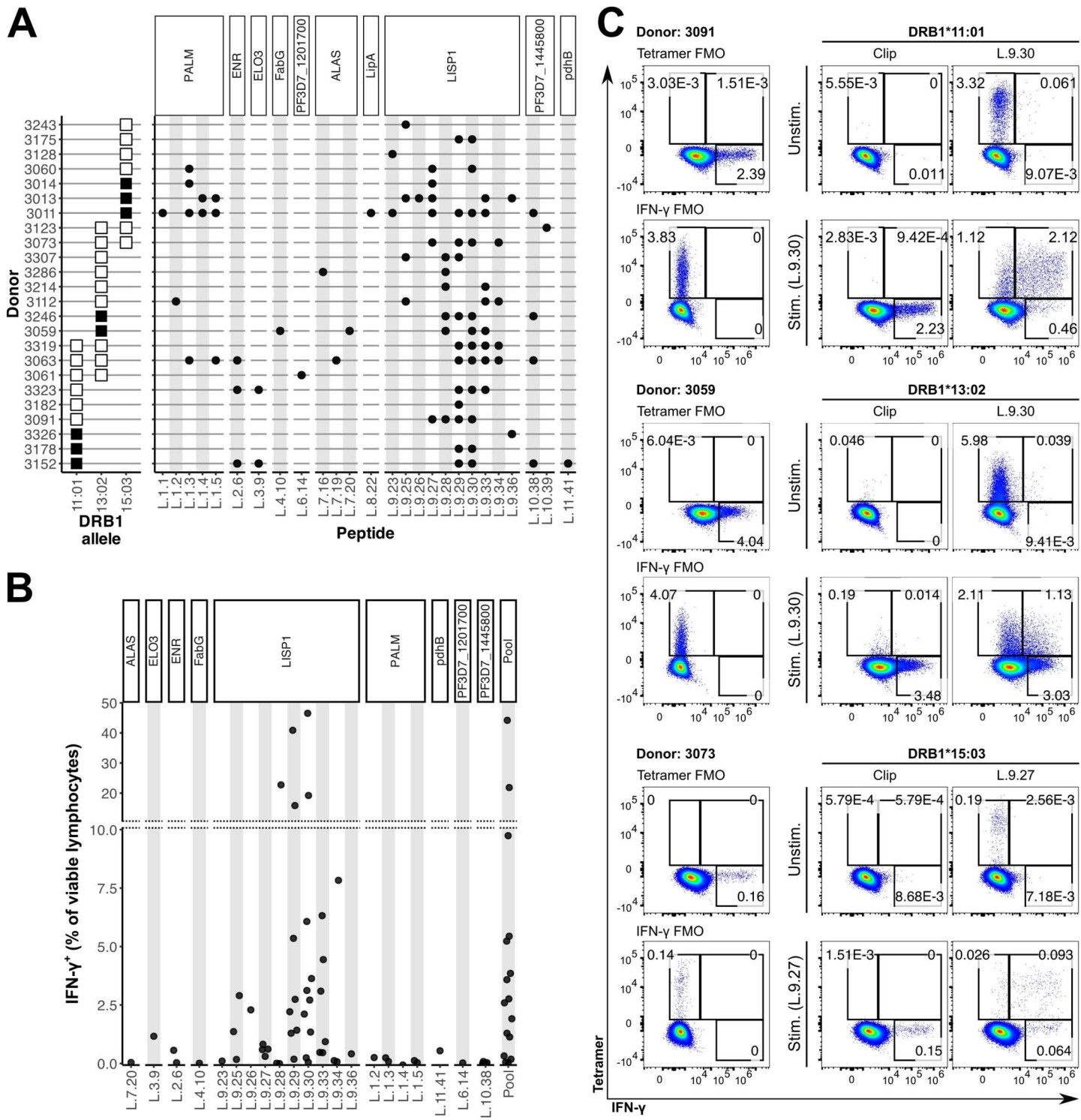

**Fig 4. CD4+ T cell responses against peptides from candidate liver stage *P. falciparum* antigens. A.** Response against individual peptides in deconvolution IFN-γ ELISPOT experiments. Left: allele haplotypes for each of the donors tested (A. donor is homozygous for DRB1 allele on the horizontal axis; A. donor carries the DRB1 on the horizontal axis and another DRB1 allele). Right: points indicate positive IFN-γ responses. **B.** Magnitude of IFN-γ response against peptides measured by flow cytometry. The gating strategy used to analyze this data is detailed in S2 Fig. In **A** and **B**, peptides (x axis) are grouped by their antigen of origin (top ribbons). **C.** pMHC tetramer staining of CD4+ T cell lines from malaria-exposed children and their IFN-γ secretion in response to stimulation with peptides L.9.30 or L.9.27. FMO: fluorescence-minus-one control.

CD4+ T cell epitopes within 10 *P. falciparum* antigens that are predominantly expressed during the liver stage of the malaria parasite life cycle. Using this approach, we were able to demonstrate *P. falciparum* epitope-specific CD4+ T cell responses in the peripheral blood of 67% of malaria-exposed Ugandan children.

Among the candidate antigens we studied, LISP1 stands out as the most broadly targeted, and it contains the epitopes that generated the highest frequency of IFN-γ producing cells in our short-term T cell lines. The *lisp1* gene has stage-selective expression, and its transcript is one of the most abundant in the mid- and late-liver stages of *P. vivax* [24,36] and *P. cynomolgi* [25]. This closely resembles the pattern of its *P. falciparum* ortholog in human fetal hepatocyte organoid cultures [38] and humanized liver mice with chimeric bone marrow [26]. *Lisp1* mRNA expression levels closely predict protein abundance in *P. yoelii* [34] and *P. berghei* [33], peaking at the late liver stage in both murine pathogens. The role of LISP1 in parasite development has been studied almost exclusively in mouse models of infection. The protein is equipped with a putative signal peptide and localizes to the parasitophorous vacuole (PV) [33,34]. In *P. yoelii*, it plays a critical role in PV rupture to allow the merozoites to egress from the liver [33]. In *P. berghei*, it is found in the intrahepatocytic and merosome proteomes [35]. Disruption of *lisp1* prevents liver egress but does not affect parasite development through the mosquito or blood stages [33]. Intriguingly, mice immunized with the LISP1 protein have reduced liver parasite burden upon infection [39]. As an immunogen, its combination with CSP improves parasite control and protection from reinfection over CSP alone by inducing liver-resident, parasite-specific CD8+ T cells [39,40]. Our results indicate that LISP1 is also a frequent target of cellular immunity in human *P. falciparum* malaria and contains epitopes that are widely recognized by CD4+ T cells.

PALM, the second most frequently recognized antigen in our study, is an apicoplast protein. Like LISP1, it is involved in merozoite formation [41], which permits the transition from the late liver to the blood stages. It is dispensable for the development of *P. berghei* sporozoites and asexual blood stages; however, *palm*−/− parasites arrest development at the late liver stage of infection. Immunization with *palm*-deleted sporozoites confers immunity from reinfection in mice [41]. Among the rest of our candidate antigens, six are enzymes involved in lipid biosynthesis pathways (ENR, ELO3, FabG, G3PDH, LipA, and pdhB), which take place mainly in the apicoplast [34,42,43]. ALAS is an enzyme that catalyzes an early, mitochondrial stage in the *de novo* heme biosynthesis pathway [44]. All of these processes are essential for the liver stage, and more specifically, late liver stage *Plasmodium* development, but can be disrupted without compromising blood stage viability. The remaining two candidates are uncharacterized conserved membrane proteins of unknown function [29]. No previous evidence of cellular immune responses against any of these antigens was found in the literature.

To identify antigens with predominant liver stage expression in *P. falciparum*, we harnessed existing datasets reporting gene expression in the liver stages of *P. cynomolgi* and *P. vivax*, and in the blood stages of *P. falciparum* and other *Plasmodium* species. While transcript abundance may not perfectly reflect protein expression levels, analyses of the dynamics of protein expression in *P. falciparum* show a generally strong correlation between the transcription of an mRNA and its translation, suggesting that at least in intraerythrocytic parasites, protein expression is primarily regulated at the transcriptional level for most genes [45–47]. We hypothesized that genes with syntenic orthology would also have higher conservation of their genomic context than non-syntenic orthologs, and that this would associate with a more conserved regulation of their expression across different *Plasmodium* species. Recent data reporting *P. falciparum* gene expression in a humanized mouse model [26] confirm that our approach qualitatively predicted preferential liver stage expression for most of the candidate antigens we selected (S1 Fig).

Although our screening covered a small percentage of the LISP1 sequence, almost two-thirds of study participants responded to at least one LISP1 peptide. A more extensive characterization of this large antigen will likely yield a greater number of relevant epitopes. Similarly, our screening covered only 3.9–12.9% of the amino acid sequence of our other candidate antigens, making them promising candidates for further exploration. In addition, proteins with similar expression patterns to that of LISP1 should be explored as potential targets of cellular immunity against liver stage malaria. Our use of stringent filters to prioritize genes with stage-selective expression may have excluded other promising antigens. For example, LISP2 violates the requirement of having syntenic orthologs in *P. falciparum*, *P. vivax,* and *P. cynomolgi*, yet it has subsequently been shown to have transcription dynamics that closely resemble those of LISP1 [26] and may be an attractive candidate for further study.

We used NetMHCIIpan for epitope prediction, as it provides robust predictive power [28] and seamless integration into our approach. While a valuable tool given the lack of empirical binding data, our prediction may have missed relevant epitopes within candidate antigens. Of note, the performance of pan-allele algorithms like NetMHCIIpan depends on the availability of training data covering a broad range of HLA alleles [48], but alleles prevalent in African populations are underrepresented in MHC ligand datasets. For example, the IEDB currently contains >10,000 epitopes from >1600 antigens that are restricted by the predominantly Caucasian allotype DRB1*07:01, in contrast with 22 epitopes from eight antigens for DRB1*15:03, which is seen almost exclusively in individuals of African descent [22,49].

While most children in this study exhibited responses against our candidate epitopes, these were not robustly detectable *ex vivo*. To overcome this, we implemented a short-term in vitro culture protocol to expand CD4+ T cells before their interrogation for a specific response. It is possible that this protocol favored the growth of epitope-specific cells that were already predisposed to proliferate (e.g., central memory cells) to the detriment of more differentiated populations with effector-like properties. As a consequence, we do not know whether the frequency of epitope-specific T cells that we measured is proportional to their frequency *in vivo*. Additionally, because we relied on *in vitro* expansion, we were unable to determine the phenotypic and functional features that these cells exhibit in the circulation of malaria-exposed children. The development of pMHC multimer reagents could facilitate a more detailed characterization of epitope-specific T cells in future studies, using sample volumes that are larger but still feasible to obtain from research participants. Similarly, using a larger number of epitopes in these antigens presented by a broader variety of relevant HLA alleles could afford enough sensitivity to detect and characterize circulating specific T cells using ex vivo assays, without prior T cell expansion.

Seeking to maximize our chances of detecting responses against liver stage antigens in circulation, we used PBMC obtained within 90 days following a documented malaria episode. In this range, responses against liver stage epitopes were most frequently observed in donors with a less recent episode of malaria. This is in agreement with results from a controlled human infection after immunization with attenuated sporozoites, in which cells secreting IFN-γ in response to pre-erythrocytic antigens were detected in a higher proportion of volunteers at 28 days after challenge, compared to 3 days after challenge [7]. Similarly, observational data in a childhood malaria cohort from Uganda suggested that T cells specific against pre-erythrocytic antigens are most prevalent in circulation between days 61–180 after a clinical malaria episode, whereas T cells responding against blood stage antigens were more frequently associated with more recent malaria [31]. This could indicate that, after an infection, there is a delay between the establishment of a liver stage-specific CD4+ T cell response

and the emergence of a population of memory T cells in the peripheral circulation that is large enough to be detected by our experimental approach.

In sum, we have identified antigens with selective expression in the liver stage of *P. falciparum* that are frequent targets of the CD4+ T cell response in chronically malaria-exposed Ugandan children. Within these antigens, we have identified several highly targeted epitopes, and in some cases the restricting MHC alleles, and validated this restriction using pMHC multimers. This work advances our understanding of the T cell response to liver stage malaria and offers important new tools for the investigation of correlates of protective immunity, which may help unlock new strategies to improve anti-malarial vaccines. Future studies could leverage a similar approach to screen a larger number of epitopes and determine the frequency and durability of CD4+ T cell responses to LISP1 and other candidate liver stage antigens following both natural infection and attenuated sporozoite vaccination strategies. Most importantly, further research is needed to identify correlations between responses against these liver stage antigens and prospective protection from malaria in larger human cohorts. This additional insight would help identify attractive candidates for inclusion in next-generation, multi-stage malaria vaccines.

## Participants, materials, and methods

### Ethics statement

This study was carried out following the recommendations of the Uganda National Council of Science and Technology and the institutional review boards of the University of California, San Francisco, and Makerere University, with written informed consent from all adult participants or parents/guardians of child study participants. All subjects gave written informed consent in accordance with the Declaration of Helsinki. The protocol was approved by the institutional review boards of the University of California, San Francisco, and Makerere University.

### Study participants

Participants in this report were enrolled in the PRISM observational study [23] in Tororo, Uganda. This area is holoendemic for malaria, with slight increases in incidence during two annual seasonal peaks (October-January and April-July). The study included quarterly clinical visits with active surveillance for malaria parasitemia and additional visits to evaluate for malaria in case of fever or other symptoms of malarial disease. Blood specimens obtained from children and adults in PRISM were processed to isolate and cryopreserve peripheral blood mononuclear cells (PBMC). A malaria event was defined as febrile illness (tympanic temperature ≥ 38 ºC) and concurrent parasitemia detected by microscopy of blood smears stained with Giemsa. A more detailed description of the parent study cohort and the PRISM study design is published elsewhere [23].

For this study, we selected specimens from a subset of 34 children who were selected on the basis of age (5–11 years), HLA-DRB1 alleles (*11:01, *13:02, and/or *15:03), and the availability of a PBMC sample that was obtained within 90 days following an episode of malaria. Participants included in this report were 44.1% female (15/34), with a median age of 7.6 (interquartile range: 6.1–9.3), and were sampled 4–83 days after their last recorded episode of malaria (median: 21.5, interquartile range: 8.25–53).

No sex-based difference was observed in the proportion of participants responding against the total pool of peptides (Fisher's exact test p=0.15) or the peptides recognized by at least four donors (p>0.24) in this study.

## Selection of candidate *Plasmodium* antigens

Transcriptomic datasets used for candidate antigen selection were obtained from PlasmoDB [29]. The datasets used included gene expression data for *P. vivax* [24,50,51], *P. cynomolgi* [25,52], *P. berghei* [53,54], and *P. falciparum* [55–59]. Protein sequences for candidate antigens were sourced from UniProt. An additional dataset reporting mRNA expression of *P. falciparum* in mice with humanized chimeric liver was extracted from the supplementary data of a preprint article [26]. Protein sequences for the candidate antigens were downloaded from the UniProt resource.

A scoring function was designed to prioritize candidate antigens with likely stage-selective gene expression. This function was composed of three factors, $F_1$, $F_2$, and $F_3$, each representing a criterion used for prioritization.

$F_1$ was constructed using the *P. vivax* and *P. cynomolgi* gene expression data to represent the discrepancy of expression between liver and blood stages. If $x_i^s$ is the expression in normalized CPM for a given gene in stage s for species i (with s being one of L for liver and B for blood, and i one of Pv for *P. vivax* or Pc for *P. cynomolgi*), the factor $F_1$ was defined for that gene as follows:

$$X_i = \log_2\left[\frac{\text{rank}\left(x_i^L\right)}{\text{rank}\left(x_i^B\right)}\right] \cdot \left|\text{rank}\left(x_i^L\right) - \text{rank}\left(x_i^B\right)\right|$$

Thus, the more stage-discrepant the expression of a given gene is in species i, the higher the magnitude of the resulting $X_i$ value. The value of $F_1$ was calculated as $F_1 = X_{Pv} + X_{Pc}$. Genes with $F_1 > 0$ have preferential liver stage expression, while $F_1 < 0$ have higher expression in blood stage parasites.

$F_2$ represents the expression of a given gene in the asexual intraerythrocytic stages of *P. falciparum*. For each gene, a value Y was calculated as the $\log_{10}$-transformed sum of normalized CPM values across time points in the intraerythrocytic cycle development transcriptomic data. Y values for all genes in the dataset were min-max-normalized to values between 0 and 1. $F_2$ was defined as 1-Y for genes with $F_1>0$, and as Y for genes with $F_1<0$. Thus, $F_2$ modulates the value of F1 to produce larger, positive values for genes with preferential expression in the liver stages of *P. vivax* and *P. cynomolgi* and larger, negative values with preferential expression in the blood stage of said species.

Finally, factor $F_3$ gives greater weight to genes with more strongly correlated expression between *P. falciparum* and other *Plasmodium* species at a given, matched stage in the life cycle.

**Table 2. Summary of datasets used for the calculation of factor $F_3$, a component of the score used to prioritize candidate antigens on the basis of their transcript expression. (IEDC: intra-erythrocytic development cycle).**

| *P. falciparum* data | Comparator species | Comparator species data | Comments |
|---|---|---|---|
| Blood stage IEDC [60] | *P. cynomolgi* | Infected macacques followed for 100 days [61] | Data was summarized as the mean across specimens for each species. |
| Asexual blood stages [62] | *P. vivax* | Blood stage IEDC [63] | Data was summarized as the mean across specimens for each species. |
| Blood stage IEDC [60,64] | *P. berghei* | Blood stage IEDC [65] | Data was summarized as the mean expression across timepoints for both species. Only "control asexual" specimens were used from the *P. berghei* dataset. |
| Enriched asexual blood stages [62] | *P. berghei* | Enriched asexual blood stages [66] | Ring, trophozoite, schizont, and gametocyte stage data was matched for each gene in both species, and correlation was calculated separately for each stage. |
| Salivary gland sporozoite [67] | *P. vivax* | Salivary gland sporozoite [68] | Cultured sporozoites data in the *P. falciparum* dataset was ignored. |

For each gene, the final score was calculated as $S = F_1 \cdot F_2 \cdot F_3$.

To calculate its value, data was subset in pairs as shown in **Table 2**, and a robust linear regression curve was fit to the data for each pair. For each gene and each inter-species comparison, a value z was calculated as the residual value of the linear fit for that gene divided by the expected value according to the fitted line. If Z is the sum of all z values for a given gene, the factor F3 was defined as $F_3 = \dfrac{1}{|Z|}$.

## *P. falciparum* epitope prediction and conservation analysis

NetMHCIIpan v. 4.1 [28] was used for epitope prediction. Starting with the four top-scoring candidate antigens, the prediction was run, and the number of strong binders for each allele was assessed. Additional antigens prioritized by their score were iteratively added until a minimum of 23 predicted strong binders (Rank% ≤ 0.5) was reached for each of the target alleles. This condition was met at 12 candidate antigens. One antigen (KASIII, PF3D7_0211400) was removed due to its complete absence of predicted strong binders for DRB1*11:01.

Overlapping predicted epitopes differing by fewer than 3 amino acids were collapsed into merged peptides up to a maximum length of 19 amino acids. This used software developed in R around tools in the seqinr [69] and stringdist [70] packages. Peptides predicted to bind to multiple alleles among our three targets were given a higher priority in the list of candidates to synthesize and test. The rest of the predicted binders were prioritized by their predicted binding score (%Rank). The final list of 41 synthesized peptides contained 23 predicted strong binders for each allele, except for DRB1*11:01, which after consolidation of overlapping sequences accounted for 17 predicted strong binders.

In order to avoid epitopes with highly variable sequences, ClustalW alignments were generated for each antigen using the protein sequences from *P. falciparum* strains deposited in PlasmoDB [29]. Conservation was then calculated for each 15-mer as the proportion of strains that had identical sequence to the reference (3D7) strain. Four LISP1 (each 67% conserved) and 3 pdhB (each 50% conserved) 15-mers were eliminated by this criterion, while all remaining 15-mers were fully conserved across strains.

Sequence conservation was further assessed in *P. falciparum* field isolates from Eastern Africa (more specifically, sequences obtained from 398 Eastern African isolates, contributed from Uganda, Kenia, Tanzania, and Malawi) deposited in the MalariaGEN initiative database [71] (v. 6). The final selection of peptides had a mean conservation of 99.8%, and minimum conservation of 95.3%.

A BLAST search was performed against the NCBI non-redundant protein database to address possible cross-reactivity with sequences from other pathogens. All predicted epitopes were identical only to *Plasmodium*-derived sequences, and were confirmed not to have duplicate occurrences in other *P. falciparum* antigens.

## Peptides and T cell expansion

All peptides were purchased from Genscript (Piscataway, NJ, USA) as lyophilized powder, reconstituted in DMSO at a concentration of 10 mg/ml, aliquoted, and preserved at -80 ºC until use.

Cryopreserved PBMC were thawed into HS-R10 medium (RPMI 1640 supplemented with 10% v/v heat-inactivated human serum, 5 mM L-Glutamine, 100 IU/ml penicillin, 0.1 mg/ml streptomycin and 10 mM HEPEs), counted, and rested for 2–3 h in the same medium before stimulation.

Between 1.0–3.5×10⁶ PBMC were split into 10 wells of a 96-well U-bottom plate and stimulated with pooled peptides (10 μg/ml total peptide, 0.26 μg/ml each peptide) for 24 h.

The stimulus was then diluted down by aspirating 150 μl/well supernatant and replacing it with fresh HS-R10 medium. On day 3, 150 μl/well of media was aspirated and replaced with an equal amount of HS-R10 supplemented with recombinant human IL-2 (IL-2; Peprotech, Cranbury, NJ, USA), to a final concentration of 50 IU/ml. Starting on day 7, every 3–4 days, cells from each donor were pooled and counted, and their density was adjusted to approximately $10^6$ cells/ml before replating them, with the simultaneous addition of IL-2 to a final concentration of 50 IU/ml. Cells were transferred to 48-, 12- and 6-well plates as necessary to accommodate their growth rate.

PBMC from three adult, malaria-naïve control donors were expanded alongside our study subject specimens to test for the unwanted expansion of irrelevant cross-reactive cells. No response against the total peptide pool or the screening matrix pools was observed for the T cells expanded from malaria-naïve controls.

### ELISPOT screening

ELISPOT screenings were performed using the ImmunoSpot human IFN-γ ELISPOT kit (CTL, Shaker Heights, OH, USA) following the protocol provided by the manufacturer. Briefly, plates were activated with 15 μl/well 70% ethanol for less than 1 min, washed with PBS, and coated with capture anti-IFN-γ antibody overnight at 4 ºC. Cultured cells were plated at a density of $7.5 \times 10^3$—$2.5 \times 10^4$ cells/well (depending on growth rate and cell availability) and stimulated with pooled peptides for 18 h. IFN-γ secretion was detected with a biotinylated anti-IFN-γ antibody, a streptavidin-alkaline phosphatase conjugate, and the CTL TrueBlue developing reagent, all provided with the kit.

Total pooled peptides ("L pool") were plated at 10 μg/ml total peptide. For matrix peptide mixes a-l, concentration was adjusted to 0.256 μg/ml per peptide to match the concentration of individual peptides in the L pool condition. For individual peptide stimulation, peptide concentration was 10 μg/ml. Each peptide stimulation condition was tested in duplicate. PHA was used at a 2 μg/ml concentration as a positive control. A negative control (lacking peptides or PHA) was set up in triplicate for each T cell line. DMSO was added as needed to equalize the solvent concentration across test and control conditions (0.1% v/v, determined by the maximum DMSO concentration reached in peptide-stimulated wells).

Spots were enumerated using an ImmunoSpot S6 analyzer (CTL). A response was considered positive if a well had > 5 spot forming units (SFU) and the mean SFU for a given condition was higher than the mean SFU plus two standard deviations of the corresponding negative controls. Unless stated otherwise, results are generally expressed as spot-forming units per $10^6$ input cells (SFUPMC).

### Flow cytometry

For stimulation assays, cells were counted and plated with the assay stimuli in HS-R10 medium, at cell culture conditions (37 ºC in a 5% $CO_2$, humidity-controlled atmosphere). All stimuli and DMSO concentrations were identical to those used in ELISPOT assays. After 3 h of incubation with the stimuli, 1 μl/well monensin (GolgiStop, BD Bioscience) was added, then cells were returned to the incubator for an additional 15 h. After this, cells were washed with PBS twice, transferred to a V-bottom 96-well plate, and stained using the antibodies and volumes indicated in **Table 3**, diluted in FACS staining buffer (RD Systems, Minneapolis, MN, USA). Surface stain antibodies were incubated for 30 min at room temperature, then washed three times with PBS. Cells were then permeabilized using the eBioscience FoxP3 staining buffer set, following the manufacturer-recommended protocol (ThermoFisher Scientific, Carlsbad, CA, USA). Intracytoplasmic staining with anti-IFN-γ antibody was incubated overnight

**Table 3. Antibodies and reagents for flow cytometry staining in ICS experiments.**

| Fluorochrome | Target | Clone | Concentration (µl/50 µl staining mix) | Staining step |
|---|---|---|---|---|
| LIVE/DEAD Blue | Viability | – | 0.1 | Surface |
| BUV737 | CD8a | RPA-T8 | 2.0 | Surface |
| FITC | CD56 | HCD56 | 1.0 | Surface |
| PE-Fire 700 | γδTCR | B1 | 2.0 | Surface |
| Alexa Fluor 700 | CD4 | RPA-T4 | 0.5 | Surface |
| APC-Cy7 | CD3 | UCHT1 | 0.5 | Surface |
| BV421 | IFN-γ | 4S.B3 | 1.0 | Intracellular |

**Table 4. Antibodies and reagents in flow cytometry panel for pMHC tetramer staining.**

| Fluorochrome | Target | Clone | Concentration | Staining step |
|---|---|---|---|---|
| PE | pMHC tetramer | N/A | 6 µg/ml (DRB1* 11:01 and *13:02 constructs) 2 µg/ml (DRB1*15:03 constructs) | Pre-stimulation |
| BUV737 | CD8a | RPA-T8 | 2.0 µl/50 µl | Surface |
| BUV805 | CD3 | UCHT1 | 2.0 µl/50 µl | Surface |
| PerCP | CD4 | RPA-T4 | 0.5 µl/50 µl | Surface |
| PE-Fire 700 | γδTCR | B1 | 2.0 µl/50 µl | Surface |
| Alexa Fluor 700 | CD19 | HIB-19 | 0.5 µl/50 µl | Surface |
| Alexa Fluor 700 | CD14 | M5E2 | 0.5 µl/50 µl | Surface |
| BV421 | IFN-γ | 4S.B3 | 1.0 µl/50 µl | Intracellular |

Events were collected on a Cytek Aurora 5L spectral flow cytometer. Data was unmixed and spillover-corrected using SpectroFlo (Cytek Biosciences, Fremont, CA) and analyzed using FCSExpress v. 7 (DeNovo Software, Pasadena, CA) or FlowJo v. 10.10 (BD Life Sciences).

at 4 ºC, then washed three times with permeabilization buffer. Stained cells were fixed with 1% PFA in PBS for 20 min at room temperature and washed once with PBS before storage at 4 ºC until collection.

Peptide-MHC tetramers were synthesized by the NIH Tetramer Core Facility at Emory University. For tetramer staining experiments, this procedure was modified to overcome the downregulation of TCR and loss of staining observed upon peptide-specific stimulation. Cells were counted, washed once with PBS, and plated in a V-bottom, 96-well plate. Tetramer staining mixes were diluted in FACS staining buffer at concentrations determined in a prior optimization assay (**Table 4**), then applied to the plated cells at 50 µl/well and incubated for 90 min at cell culture conditions. At the end of this incubation, 150 µl/well PBS was added to wash excess tetramer, then cells were re-suspended in HS-R10 medium and transferred to a U-bottom plate containing the assay stimuli. After 3 h of incubation in cell culture conditions, 1 µl/well monensin (GolgiStop, BD Bioscience) was added, then cells were returned to the incubator for an additional 15 h. Cells were then incubated with the surface stain panel detailed in **Table 4** for 1 h at 4 ºC. The staining mix was then washed three times with PBS and staining proceeded for intracytoplasmic IFN-γ as described above.

## Supporting information

**S1 Fig. Validation of stage-selective transcription of our chosen candidate antigen genes in *P. falciparum*. A.** Gene expression in the pre-erythrocytic stages of *P. falciparum* (sporozoites obtained from *Anopheles* mosquito salivary glands and intrahepatocytic parasites obtained from humanized liver mice), as reported by Zanghi et al. [26]. **B.** Cumulative expression

of genes in the blood and liver stages of *P. falciparum* derived from the datasets reported in Zanghi et al. [26] and Kucharski et al. [58]. Each point is a gene. Genes selected as candidate antigens are highlighted in color in **A** and **B**.
(TIFF)

**S2 Fig. Flow cytometry analysis of expanded T cells responding to liver stage *P. falciparum* epitopes. A.** Gating strategy used to analyze flow cytometry data (one representative T cell line and stimulation condition shown). **B.** Distribution into T cell subsets of IFN-γ+ lymphocytes responding against stimulation with the total pool of candidate epitopes. Each data point represents a T cell line. Populations were defined by boolean combinations of the IFN-γ+ gate and the T cell subset gates as indicated by dashed arrows in A. **C.** IFN-γ production by CD4+ αβ T cells (gated upstream in lymphocytes/Single cells/Viable cells/ CD3+CD56−/γδTCR−), after 18 h of culture in unstimulated condition or upon stimulation with the total pool of liver stage peptides, or peptide L.9.30 (cells from one representative donor are shown).
(TIFF)

**S3 Fig. Screening of *P. falciparum* liver stage antigen peptides by the 2D peptide matrix approach. A.** 2D peptide matrix used for first screening step by IFN-γ ELISPOT. A pool of peptides was created for each of the columns and each of the rows of the matrix. Short-term T cell lines were stimulated in IFN-γ ELISPOT experiments using each of these pools. **B.** Result obtained from one T cell line in the ELISPOT experiment described. The pools that elicited a response are highlighted in the matrix representation on panel A. **C.** Following the results from B, single peptides were identified at the intersections highlighted in A and tested individually in the second screening step (deconvolution). The resulting measured response pinpointed L.9.29 and L.9.30 as targets of the T cell line under study. **D.** Correlation between IFN-γ ELISPOT SFUPMC (day 17 of T cell culture) and frequency of IFN-γ+ events observed by flow cytometry (day 28 of T cell culture) for short-term T cell lines stimulated with individual peptides or the total pool of candidates. Line and error band represent adjusted robust linear regression. Spearman's $\rho$ and p-value are displayed. In **B** and **C**, numbers indicate the mean SFUPMC from duplicate-assayed wells for peptide-stimulated conditions, and triplicate negative control wells (Neg). Error bars show the maximum and minimum SFUPMC for each condition. SFUPMC: spot-forming units per $10^6$ input cells.
(TIFF)

**S4 Fig. Upstream gating for tetramer staining data.** Flow cytometry gating strategy used to analyze pMHC multimer staining on primary T cell lines. Representative plots for one T cell line are shown.
(TIFF)

## Acknowledgments

The authors would like to thank the study participants and their families, and the study team members for their efforts and contributions. We are also grateful to Drs. Gina Borgo, Paul Ogongo, Josephine Reijneveld, Rachel Rutishauser, Sara Suliman and Joel Ernst for their valuable insight and advice; to the Core Immunology Lab for their assistance with flow cytometry equipment; to the NIH Tetramer Core Facility at Emory University (contract number 75N93020D00005) for their production of the pMHC tetramers featured in this work; and to the VEuPathDB/PlasmoDB and MalariaGEN DB teams for their remarkable work, without which this project would not have been possible.

## Author contributions

**Conceptualization:** Gonzalo R. Acevedo, Margaret E Feeney.

**Data curation:** Gonzalo R. Acevedo, John Rek, Grant Dorsey.

**Formal analysis:** Gonzalo R. Acevedo.

**Funding acquisition:** Gonzalo R. Acevedo, John Rek, Grant Dorsey, Margaret E Feeney.

**Investigation:** Gonzalo R. Acevedo, Sophie S. Samiee, Mikias Ilala, Justine Levan, Meagan E. Olive, Riana D. Hunter, Raja Rajalingam.

**Methodology:** Gonzalo R. Acevedo, Mary Prahl.

**Software:** Gonzalo R. Acevedo.

**Supervision:** Gonzalo R. Acevedo, Raja Rajalingam, Margaret E Feeney.

**Visualization:** Gonzalo R. Acevedo.

**Writing – original draft:** Gonzalo R. Acevedo, Margaret E Feeney.

**Writing – review & editing:** Gonzalo R. Acevedo, Sophie S. Samiee, Mikias Ilala, Justine Levan, Meagan E. Olive, Riana D. Hunter, Mary Prahl, Raja Rajalingam, John Rek, Grant Dorsey, Margaret E Feeney.

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
