## [Decision Letter · Decision Letter 0]

24 Nov 2024

PPATHOGENS-D-24-02133Liver stage *P. falciparum* antigens highly targeted by CD4^+^ T cells in malaria-exposed Ugandan childrenPLOS Pathogens Dear Dr. Feeney, Thank you for submitting your manuscript to PLOS Pathogens. After careful consideration, we feel that it has merit but does not fully meet PLOS Pathogens's publication criteria as it currently stands. Therefore, we invite you to submit a revised version of the manuscript that addresses the points raised during the review process. Please submit your revised manuscript within 60 days Jan 23 2025 11:59PM. If you will need more time than this to complete your revisions, please reply to this message or contact the journal office at plospathogens@plos.org. Please include the following items when submitting your revised manuscript:* A rebuttal letter that responds to each point raised by the editor and reviewer(s). You should upload this letter as a separate file labeled 'Response to Reviewers'. This file does not need to include responses to any formatting updates and technical items listed in the 'Journal Requirements' section below.* A marked-up copy of your manuscript that highlights changes made to the original version. You should upload this as a separate file labeled 'Revised Manuscript with Track Changes'.* An unmarked version of your revised paper without tracked changes. You should upload this as a separate file labeled 'Manuscript'. If you would like to make changes to your financial disclosure, competing interests statement, or data availability statement, please make these updates within the submission form at the time of resubmission. Guidelines for resubmitting your figure files are available below the reviewer comments at the end of this letter. We look forward to receiving your revised manuscript. Kind regards, Tracey J. LambSection EditorPLOS Pathogens Tracey LambSection EditorPLOS Pathogens Michael Malim

Editor-in-Chief

PLOS Pathogens

orcid.org/0000-0002-7699-2064 **Journal Requirements:** 1) We note that your Data Availability Statement is currently as follows: " All relevant data are within the manuscript and its Supporting Information files". Please confirm at this time whether or not your submission contains all raw data required to replicate the results of your study. Authors must share the “minimal data set” for their submission. PLOS defines the minimal data set to consist of the data required to replicate all study findings reported in the article, as well as related metadata and methods (https://journals.plos.org/plosone/s/data-availability#loc-minimal-data-set-definition). For example, authors should submit the following data: - The values behind the means, standard deviations and other measures reported; - The values used to build graphs; - The points extracted from images for analysis. Authors do not need to submit their entire data set if only a portion of the data was used in the reported study. If your submission does not contain these data, please either upload them as Supporting Information files or deposit them to a stable, public repository and provide us with the relevant URLs, DOIs, or accession numbers. For a list of recommended repositories, please see https://journals.plos.org/plosone/s/recommended-repositories. If there are ethical or legal restrictions on sharing a de-identified data set, please explain them in detail (e.g., data contain potentially sensitive information, data are owned by a third-party organization, etc.) and who has imposed them (e.g., an ethics committee). Please also provide contact information for a data access committee, ethics committee, or other institutional body to which data requests may be sent. If data are owned by a third party, please indicate how others may request data access. **Reviewers' Comments:** Reviewer's Responses to Questions

**Part I - Summary**

Reviewer #1: The manuscript by Acevedo et al. addresses a critical gap in knowledge regarding T cell responses to the P. falciparum liver stage, which hinders the identification of targets for protective immunity and effector functions and thus posing a hurdle to the development of improved vaccines.

This is an interesting study that reports a strategy and tools combining bioinformatics and immunological assays to characterize human T cell immunity against the liver stage. The authors identify an antigen (LISP1) that is frequently targeted in Ugandan children. Therefore, the manuscript will be of significant interest to the malaria immunology and vaccinology community. Additionally, the manuscript is clear and well-written.

Reviewer #2: Acevedo et al use bioinformatics and existing transcriptomic data to generate a list of genes with suggested liver-stage specific expression from P. falciparum. They then down selected further to 11 candidate proteins that were assessed for predicted epitopes for 3 MHC II alleles expressed in children from a study region of Uganda. After amplification of PBMC in vitro with peptide pools, responses were detected by ELISPOT and single epitopes deconvoluted and tested by IFNg production and MHC II tetramer binding. The results were clear, most children responded to at least one antigen, with the parasitophorous vacuole localized LISP1 evoking the most frequent and largest responses. The work appears to be scientifically rigorous and the data are clearly presented. The identification of new CD4 T cell epitopes from liver-stage P. falciparum is welcome.

Reviewer #3: Due to the importance of CD4 T cells in helping generate antibody responses to Plasmodium, Acevedo et al. sought to identify and investigate liver stage P. falciparum antigens targeted by CD4 T cells. Utilizing existing datasets and bioinformatic tools, the authors interrogate genes that are expressed during liver stage and used netMHCII to predict MHCII-binding epitopes. They tested these peptides in long-term stimulation assays and found that most malaria-exposed Ugandan children had detectable CD4 T cell responses to one or more antigens. Finally, the authors generated peptide:MHC tetramers to identify antigen-specific CD4 T cells. This well-written manuscript provides important information (identification of CD4 T cell epitopes/peptides to liver stage antigens) and tools (pMHC tetramers, cell lines). Overall, the experiments are well-executed, and the data used to support the conclusions of the paper are solid.

**Part II – Major Issues: Key Experiments Required for Acceptance**

Reviewer #1: One of my main concerns, and a limitation of the study, is the assumption that the identified antigens are targets of protective T cell immunity. For example, in the abstract, it is stated that “those novel liver-stage P. falciparum epitopes represent promising new vaccine targets." However, the fact that T cell responses were detected in children from a highly exposed area with a recent clinical episode suggests that these responses are not protective and more likely reflect recent exposure. This point has not been discussed in the manuscript. Is there any data on subsequent malaria episodes or infections in these children? If so, correlates of protection analyses could be performed, although the sample size is limited. Alternatively, could samples from children without malaria (i.e., protected individuals) be analyzed?

Not clear if all peptides were tested in non-exposed negative controls for cross-reactive responses.

In Figure 4, donor 3059, the gating of IFN-g does not look ok for CLIP and L.9.30, the gate seems to cross the IFN-g negative populations (both the tetramer negative and positive).

Reviewer #2: Major concern

1. Given the high response frequency observed here after in vitro amplification but lack of detectable response direct ex vivo, this reviewer would like to see a negative control where PBMC from malaria naïve donors are in vitro amplified-specifically with the LISP1 epitope pools and tested in the ELISPOT

Reviewer #3: Please define Plasmodium-specific antibody titers in these children, especially the titers for LISP1-specific antibodies. This outcome measure could provide context for the amount of “help” the LISP1-specific T cells are providing to the LISP1-specific B cells.

To establish clinical significance of the epitopes defined in this report, it would be of interest to know follow up information about these children in terms of future malaria episodes in a given 3-12 month time period. It is tempting to hypothesize that children with more Plasmodium-specific CD4 T cells would have fewer malaria episodes. While correlation does not mean causation, this would be of interest if the information is available for the cohort of children.

**Part III – Minor Issues: Editorial and Data Presentation Modifications**

Reviewer #1: The statement that the efficacy of RTS,S is lower in populations with greater malaria exposure is inaccurate. The reference provided also makes this assumption without enough evidence. In fact, there is no clear evidence that higher exposure is associated with lower efficacy or protection, with some studies showing a trend or a small effect, while others show no effect or even a correlation between higher CSP or malaria antibodies at baseline with greater immunogenicity, which is linked to higher protection. The endpoints in clinical trials are often confounded by factors related to malaria incidence, making it difficult to isolate the effects of transmission intensity. This statement should be revised or removed.

It is not clear how the 41 peptides were shortlisted from the 295 predicted peptides.

Why were children selected for the study instead of adults, who would have had longer exposure and potentially higher immunity?

Some responders in the EliSpot assay had no detectable responses in the ICS assay. However, for those with positive responses in both assays, the correlation was quite good. Why do the authors believe that the ICS assay showed less sensitivity in some cases?

Lastly, in line 152, there is an error with a date and time inserted.

Reviewer #2: Minor concerns

1. Please check carefully for typos in text and references

2. Some discussion is warranted as to how the authors would proceed to prioritize these antigens for vaccine assessment

3. Some discussion is warranted as to how the authors would assess the effectiveness of vaccination against one or more prioritized antigens

4. It is not exactly clear what criteria were used to downselect to the 11 candidate antigens

Reviewer #3: Is it possible to perform tetramer enrichment on PBMCs from individual or pooled patients to begin to identify surface phenotypes of these antigen-specific CD4 T cells? HLA typing and blood volumes may be a limiting factor with this experiment.

Typo on lines 151-152 (date/time inserted instead of a reference?)

PLOS authors have the option to publish the peer review history of their article (what does this mean?). If published, this will include your full peer review and any attached files.

Reviewer #1: No

Reviewer #2: No

Reviewer #3: **Yes: **Kristina Burrack

---

## [Decision Letter · Decision Letter 1]

27 Jan 2025

Dear Dr. Feeney,

We are pleased to inform you that your manuscript 'Liver stage *P. falciparum* antigens highly targeted by CD4^+^ T cells in malaria-exposed Ugandan children' has been provisionally accepted for publication in PLOS Pathogens.

Best regards,

Tracey J. Lamb

Section Editor

PLOS Pathogens

Tracey Lamb

Section Editor

PLOS Pathogens

Sumita Bhaduri-McIntosh

Editor-in-Chief

PLOS Pathogens

orcid.org/0000-0003-2946-9497

Michael Malim

Editor-in-Chief

PLOS Pathogens

orcid.org/0000-0002-7699-2064

Reviewer Comments (if any, and for reference):

Reviewer's Responses to Questions

**Part I - Summary**

Reviewer #1: The authors have addressed all my concerns and I have no further comments.

Reviewer #2: The authors addressed my concerns

Reviewer #3: Due to the importance of CD4 T cells in helping generate antibody responses to Plasmodium, Acevedo et al. sought to identify and investigate liver stage P. falciparum antigens targeted by CD4 T cells. Utilizing existing datasets and bioinformatic tools, the authors interrogate genes that are expressed during liver stage and used netMHCII to predict MHCII-binding epitopes. They tested these peptides in long-term stimulation assays and found that most malaria-exposed Ugandan children had detectable CD4 T cell responses to one or more antigens. Finally, the authors generated peptide:MHC tetramers to identify antigen-specific CD4 T cells. This well-written manuscript provides important information (identification of CD4 T cell epitopes/peptides to liver stage antigens) and tools (pMHC tetramers, cell lines). Overall, the experiments are well-executed, and the data used to support the conclusions of the paper are solid. The authors addressed this reviewer's concerns and comments, and the manuscript has been appropriately modified.

**Part II – Major Issues: Key Experiments Required for Acceptance**

Reviewer #1: (No Response)

Reviewer #2: (No Response)

Reviewer #3: No major issues.

**Part III – Minor Issues: Editorial and Data Presentation Modifications**

Reviewer #1: (No Response)

Reviewer #2: (No Response)

Reviewer #3: It may be useful to comment in the manuscript that the study was underpowered to assess association with LS pool responders vs non-responders and protection from a subsequent malaria episode. This may increase transparency as other readers will likely be interested in that outcome.

PLOS authors have the option to publish the peer review history of their article (what does this mean?). If published, this will include your full peer review and any attached files.

Reviewer #1: No

Reviewer #2: No

Reviewer #3: **Yes: **Kristina S. Burrack

---

## [Editor Report · Acceptance letter]

Dear Dr. Feeney,

We are delighted to inform you that your manuscript, "Liver stage *P. falciparum* antigens highly targeted by CD4^+^ T cells in malaria-exposed Ugandan children," has been formally accepted for publication in PLOS Pathogens.

Best regards,

Sumita Bhaduri-McIntosh

Editor-in-Chief

PLOS Pathogens

orcid.org/0000-0003-2946-9497

Michael Malim

Editor-in-Chief

PLOS Pathogens

orcid.org/0000-0002-7699-2064